# Determination of Antidepressants and Antipsychotics in Dried Blood Spots (DBSs) Collected from Post-Mortem Samples and Evaluation of the Stability over a Three-Month Period

**DOI:** 10.3390/molecules24203636

**Published:** 2019-10-09

**Authors:** Matteo Moretti, Francesca Freni, Beatrice Valentini, Claudia Vignali, Angelo Groppi, Silvia Damiana Visonà, Antonio Marco Maria Osculati, Luca Morini

**Affiliations:** Department of Public Health, Experimental and Forensic Medicine, University of Pavia, via Forlanini 12, 27100 Pavia, Italy; frafre93@gmail.com (F.F.); beatricevalentini2806@gmail.com (B.V.); claudia.vignali@unipv.it (C.V.); angelo.groppi@unipv.it (A.G.); visona.silvia@gmail.com (S.D.V.); antonio.osculati@unipv.it (A.M.M.O.); luca.morini@unipv.it (L.M.)

**Keywords:** dried blood spot, antidepressants, antipsychotics, LC-MS/MS, post-mortem, stability

## Abstract

An LC-MS/MS method for the identification and quantification of antidepressants and antipsychotics was developed on dried blood spots (DBSs). Moreover, analyte stability on DBSs within a 3-month period was monitored. Aliquots of 85 µL of blood from autopsy cases were pipetted onto DBS cards, which were dried and stored at room temperature. DBSs were analyzed in triplicate immediately, within the following 3 weeks, and after 3 months. For each analysis, a whole blood stain was extracted in phosphate buffer and purified using Solid Phase Extraction (SPE) cartridges in order to avoid matrix effects and injected in the LC-MS/MS system. Thirty-nine molecules were screened. Limits of detection (LODs) ranged between 0.1 and 3.2 ng/mL (g) and 0.1 and 5.2 ng/mL (g) for antidepressants and antipsychotics, respectively. Limits of quantification (LOQs) varied from 5 to 10.0 ng/mL for both. Sixteen cases among the 60 analyzed resulted positive for 17 different analytes; for 14 of these the method was fully validated. A general good agreement between the concentrations on DBSs and those measured in conventional blood samples (collected concurrently and stored at −20 °C) was observed. The degradation/enhancement percentage for most of the substances was lower than 20% within the 3-month period. Our results, obtained from real post-mortem cases, suggest that DBSs can be used for routine sample storage.

## 1. Introduction

Antidepressants (ADs) are commonly used in the treatment of anxiety and depression and represent one of the most frequently prescribed drug classes [1,2,3,4]. The increase in use of selective serotonin reuptake inhibitors has resulted in a reduction in AD overdose mortality and morbidity [5,6], but there are new issues of safety and tolerability [7,8]. 

Antipsychotics (APs) are frequently prescribed as well, especially in Western countries. [9]. Typical AP drugs (such as chlorpromazine, fluphenazine, and haloperidol) are generally indicated for the treatment of psychotic episodes, delusional disorders, and severe agitation [10], while psychoactive drugs such as risperidone, aripiprazole, quetiapine, olanzapine, and ziprasidone are classified as second-generation AP medications or “atypical antipsychotics”. Even if the exact mechanism of action is not yet completely clear, these atypical AP treatments work similarly to first-generation medicines by blocking dopamine pathways, but present a lower risk of causing extrapyramidal adverse effects [11]. For this reason, APs are increasingly being prescribed to treat sleeping disorders as well as anxiety and mood disorders, with an associated increase in AP overdoses [9,12,13,14,15,16].

In post-mortem samples from forensic toxicology cases, it is not infrequent to detect ADs [17,18] and/or APs [19,20,21], but making the diagnosis of fatal intoxication is a challenging task, mainly because the reference information about some substances is scarce or not available [21], and because the potential synergic effects of the different substances found in biological matrices still need to be clarified.

Biological fluid storage in post-mortem cases represents an important issue due to the fact that factors such as temperature, light, and humidity could influence the stability of substances of toxicological interest. 

The dried blood spot (DBS) technique is a simple and easy method that allows the sampling of few blood drops on a paper substrate, with the advantages of allowing easier transfer, simpler storage, and a smaller blood volume [22,23,24] collected through a less invasive sampling method.

This technique was initially applied for the newborn screening programs [25,26], but the number of applications has been steadily increasing, including metabolic-endocrine diagnosis, therapeutic drug monitoring, and toxicological, serological, and molecular biology studies [24,27,28,29,30,31,32,33,34,35].

In the field of forensic toxicology, the DBS assay has been developed for the identification and quantification of drugs of abuse [36,37,38,39] and psychoactive substances [40,41], including ADs [32,42,43,44] and APs [45,46,47].

However, to the best of our knowledge, there is a lack of information regarding the potential use of DBS methods for the analysis of ADs and APs on real post-mortem samples. 

Our team has already worked with DBSs collected from autopsy cases and has previously developed and validated liquid chromatographic tandem mass spectrometric (LC-MS/MS) methods for the identification and determination of cocaine and its main metabolites [48] and benzodiazepines and their metabolites [49] in these matrices. 

In this paper, we present an LC-MS/MS method developed for the identification of 22 antidepressants and 19 antipsychotics in DBSs and in whole blood collected from real post-mortem samples. 

The diagnostic reliability of DBSs vs. routine blood analyses of these substances has been evaluated, and the stability of the analytes on DBSs within a 3-month period of storage at room temperature has been assessed.

## 2. Results

Despite the low amount of sample, the method assessed to be sensitive and specific for all the analytes. All the LODs and the LOQs are listed in Table 1. 

The analytical procedure was fully validated only for compounds detected on real positive samples. Three substances (paroxetine, pimozide and dixyrazine) did not fulfill the acceptance criteria for the validation.

The method was found to be linear over the calibration range. The coefficients of determination (r^2^), calculated for the curves, were higher than 0.99. Accuracy and precisions were calculated at four (or five, according to LOQ) quality control levels. Results are listed in Table 2. 

The percentages were within the acceptance range suggested by international guidelines [50,51]. 

Recovery and matrix effects results are reported in Table 3. 

The use of Bond Elut Certify I cartridges for the SPE procedure guaranteed a good recovery, except for paliperidone, quetiapine, and fluoxetine, for which relatively low recovery percentages were observed. The reason could be due to a low affinity of these three molecules to the stationary phase of SPE cartridges. Matrix effects were found to be negligible for all the analytes.

Carry-over effects were not detected on blank samples injected after blood samples fortified at the concentration of 1000 ng/mL. 

In the period between January 2017 and September 2018, 60 post-mortem blood samples were collected. The qualitative results were in good agreement with the data obtained through routine analysis by means of GC-MS: a total of 16 cases provided positive results for at least one substance.

Three cases were positive for quetiapine (84.1–2309.7 ng/mL), venlafaxine (25.6–362.5 ng/mL), and desvenlafaxine (479.5–686.1 ng/mL). Two cases were positive for citalopram (56.9–408.3 ng/mL) and trazodone (91.1–162.0 ng/mL). Paliperidone (17.6 ng/mL), amisulpride (527.2 ng/mL), clotiapine (24.5 ng/mL), dibenzepin (483.4 ng/mL), dixyrazine (126.4 ng/mL), fluoxetine (5.6 ng/mL), fluvoxamine (2366.0 ng/mL), haloperidol (19.0 ng/mL), mirtazapine (133.7 ng/mL), N-desmethylmirtazapine (112.1 ng/mL), paroxetine (671.6 ng/mL), and pimozide (13.1 ng/mL) were found only in a single case. The analytical results of the positive cases are summarized in Table 4, together with the initial concentrations (at T0) of the analytes on DBSs.

Regarding concentrations, we observed a good qualitative and quantitative agreement between those measured on DBSs stored at room temperature at T0 analysis and blood samples stored at −20 °C, except in case 2. In fact, concentrations of fluvoxamine in DBSs were considerably lower than those measured in whole blood. 

The blood and DBS concentrations were significantly correlated (Spearman’s r_s_ 0.82999). 

Least-squares regression analyses also demonstrated a significant correlation between blood and DBS concentrations (*p* < 0.05) with r^2^ = 0.9323; equations of the fitted lines were: y = 1.6774x − 99.192 (see Figure 1).

The Bland–Altman plot, performed on the entire cohort of analytes, assessed a good agreement between the two measurements; the values exceeding the upper limit were excluded from the group of samples (see Figure 2). The plot assessed a good agreement between the two measurements.

Stability was evaluated on the 16 positive DBS samples collected. The measurements of the three replicates, for each sample, were always within the 15% of variability. The results obtained are reported in Figure 3, Figure 4, Figure 5, Figure 6, Figure 7, Figure 8, Figure 9, Figure 10, Figure 11, Figure 12, Figure 13, Figure 14, Figure 15, Figure 16, Figure 17, Figure 18 and Figure 19.

A variation higher than 20% of the initial drug concentration was considered as an indicator of possible instability.

## 3. Discussion

Forensic toxicological analysis could also be requested months after death, making it essential to choose the proper way to collect and store biological fluids. Fresh blood tends to be stored in plastic tubes at −20 °C for up to several years, usually ensuring a good stability of the analytes. However, this kind of storage requires adequate freezing rooms with high maintenance costs, besides the fact that consecutive freeze–thaw cycles of the same samples could influence the stability of the substances themselves [52], even if several ADs and APs did not show significant degradation in whole blood or plasma during repeated freeze and thaw cycles [53,54,55]. Thus, we are trying to find an alternative system that allows an easy blood collection and a simple and economic storage of the samples.

In this study, we have focused our attention on antidepressants and antipsychotics, which are drugs frequently detected in post-mortem samples from forensic toxicology cases [17,18,21]. We have developed a method for detection and quantitation of ADs and APs in dried blood spotted on a paper substrate and in blood stored at −20 °C and submitted to a freeze and thaw cycle during each analytical session. We applied a screening procedure to 39 molecules, evaluating sensitivity, specificity, selectivity, and carryover effects. Then, we validated the method for the analytes detected in real post-mortem samples (17 analytes have been identified and 14 of them have been fully validated). The analytical procedure assessed to be simple, sensitive and specific. 60 post-mortem cases have been included in the study. Sixteen samples were positive for at least one substance. The identification of analytes on DBSs and blood samples using the new developed method exactly matched with the results obtained through routine analyses by means of GC-MS procedures, which are applied to higher amounts of blood samples (typically 1 mL). No false negatives or no false positives were observed.

### 3.1. Pre-Analytical Issues 

The hematocrit is one of the most important parameters that could influence the diffusion of blood on DBS cards (and so the size of the spot, the drying time, the homogeneity, etc.): this might adversely affect the robustness and reproducibility of the analysis [56,57]. In order to overcome this problem, as described previously, we decided to analyze the whole blood spot deposed on the paper substrate after deposing an exact amount of blood on the card.

Sometimes it was difficult to apply post-mortem blood on the paper because of the presence of clots. However, most of the samples guaranteed a satisfactory deposition. The remaining samples (about 5%) were excluded from the study.

Other possible parameters that could influence the quality of the samples (such as post-mortem interval, cause of death…) have not been considered for this part of the study, because we wanted to evaluate the reliability of the method for samples collected in real cases, like those that are sent daily to our laboratory. 

### 3.2. Analytical Issues

The extraction procedure by means of mixed-mode SPE cartridges (C8 and strong cation exchanger) resulted in an excellent recovery for almost all analytes and a relatively low recovery percentage for paliperidone, quetiapine, and fluoxetine (see Table 5). 

In the present study, we used a sample preparation and an extraction procedure that is not specific for ADs and APs. Though a more specific SPE procedure could improve the efficiency, we decided not to change it in order to achieve the best compromise and extract as many compounds as possible using the same procedure. Indeed, thanks to the developed procedure it is possible to extract from blood also many other analytes of forensic interest, as already published in previous papers [48,58,59].

Another potential source of error during the sample treatment procedure is the desorption of the analytes from the paper. A ten-minute sonication guarantees a satisfying recovery of the analytes into the buffer solution. Figure 1 reports the correlation between concentrations measured in blood versus concentrations measured in DBS. Except for fluvoxamine, a general good correlation was observed, but the concentrations in DBSs are about 10–15% lower than the same measured in blood for almost all the positive samples. This could be due to an incomplete desorption of the analyte from the filter. For the further development of our study we will evaluate an increasing of sonication time.

Concentrations of fluvoxamine in DBSs were considerably lower than those measured in blood samples. This is probably due to unidentified extraction problems on real samples, even if no issues were observed in the validation phases.

The accuracy, precision, and matrix effects obtained for paroxetine, pimozide, and dixyrazine did not satisfy the acceptance criteria for validation. However, we decided to evaluate the stability also for these three substances. 

### 3.3. Stability Study

The stability of all the 17 compounds in the 16 positive cases (a total number of 25 analytes) was monitored over a three-month period on DBSs. 

It is well known that time period and temperature of storage of biological samples can have a significant influence on the stability of analytes.

We decided to monitor the stability of ADs and APs on DBSs stored at room temperature for different reasons. Firstly, all the samples were collected from real cases and the amount of blood that can be used for research purposes was limited. Therefore, we could not store a large number of blood samples at different temperatures, or lay many drops on DBSs. 

Finally, since DBS cards do not require any particular storage conditions and could be easily kept along with the documentation of the case, we aimed to evaluate if DBS testing could provide reproducible results in case of deferred analysis.

Several publications are available about the long-term stability of most antidepressants in plasma and serum, and all state that most of antidepressants are stable over a long period at −20 °C [60,61,62,63,64,65,66,67,68,69]. Several tricyclic antidepressants seem to be stable also in serum kept for 7 days at room temperature [70]. However, data on the stability of these compounds in whole blood samples [67,69] are limited or not at all available, especially regarding real post-mortem samples [71,72,73] and DBSs [42,44].

Previous studies have investigated the stability of common antipsychotics in different matrices stored at different temperatures for different periods [52,54,60,74,75]. Saar et al. [52] detected extensive degradation after 20 weeks in spiked post-mortem whole blood (with approximately 80% drug loss) in samples stored at 20 °C and 4 °C; some samples were also seriously affected by degradation (up to 50%) even when stored at −20 °C.

Comparison between stability results observed in this study and those ones obtained from previously published papers are summarized in Table 5.

The major weakness of this study is represented by the limited number of positive samples. Indeed, although the detection of ADs and APs in post-mortem samples in our Department is not infrequent, each case involves different analytes, making it difficult to collect a significant number of positive cases for each analyte.

## 4. Materials and Methods

### 4.1. Reagents and Chemicals

Clotiapine, clozapine-D4, citalopram-D6, desvenlafaxine, haloperidol, mirtazapine, N-desmethylmirtazapine, quetiapine, venlafaxine, quetiapine-D8, and formic acid for mass spectrometry were all purchased from Sigma-Aldrich (Milan, Italy). Amisulpride and clozapine were obtained by Sandoz (Sandoz Industrial Products, Trento, Italy). Amitriptyline, asenapine, and mianserin were purchased by Merck & Co. (MSD Italia, Pavia, Italy). Desipramine, duloxetine, hydroxyzine, levomepromazine, olanzapine, paliperidone, pimozide, and trazodone were obtained by LGC Standard (Milan, Italy). Chlorpromazine and fluoxetine were purchased by Lusofarmaco (Gruppo Menarini, Milan, Italy). Citalopram, fluphenazine, and nortriptyline were obtained by Lundbeck (Lundbeck Italia SPA, Milan, Italy). Dibenzepin, maprotiline, and trimipramine were purchased by Novartis (Novartis Farma, Varese, Italy). Dixyrazine was obtained by Laboratorio Farmaceutico S.I.T. (Pavia, Italy). Dothiepin and tiapride were purchased by Teofarma (Pavia, Italy). Fluvoxamine was obtained by Duphar B.V. (Solvay, Weesp, Holland). Paroxetine was purchased by SmithKline Beecham Pharmaceuticals (GSK, Verona, Italy). Promazine was obtained by Pierrel (Pierrel SPA, Milan, Italy). Reboxetine, sertraline, and ziprasidone were purchased by Pfizer Roerig (Pfizer SPA, Milan, Italy). Promethazine was obtained by Roussel-Maestretti (Milan, Italy). Protryptiline was purchased by M.S.D. S.p.a. (Milan, Italy). HPLC-grade methanol and acetonitrile were purchased from Mallinckrodt Baker (Milan, Italy). The mobile phase consisted of an aqueous solution with 0.1% (*v*/*v*) formic acid (A) and acetonitrile with 0.1% (*v*/*v*) formic acid (B). The aqueous solution was purified with a polytetrafluoroethylene (PTFE) filter of 0.2 mm (SUN SRi, Duluth, GA, USA) sonicated before using for 20 s in acetonitrile. 

Cards for DBSs (five-spot cards, Whatman 903TM) were purchased from Sigma-Aldrich (Milan, Italy). Bond Elut Certify I solid phase extraction cartridges (SPE, 200 mg) were obtained from Agilent (Milan, Italy).

### 4.2. Instrumentation and Chromatographic Conditions

LC-MS/MS analyses were performed with an Agilent 1100-1200 Series system (Agilent Technologies, Palo Alto, CA, USA) coupled with a 4000 QTRAP (Applied Biosystems/MDS SCIEX, Foster City, CA, USA) with an electrospray (ESI) Turbo V™ Ion Source. The LC instrumentation was composed of a binary pump, an isocratic pump, and an autosampler maintained at room temperature during analysis. The injector needle was externally washed with methanol prior to any injection. A Kinetex C18 column (100 × 2.1 mm i.d., 5 µm particle size) (Phenomenex, Castelmaggiore, BO, Italy) was kept at 45 °C during the analysis.

Chromatographic conditions as well as ion source parameters are listed in Table 6. 

The mass spectrometer was operating in Multiple Reaction Monitoring (MRM) mode in positive polarization. Table 7 lists the transitions chosen for all the analytes, optimized in previously published studies [85,86].

To guarantee the best sensitivity, the MRM transitions were divided into two groups and each sample was injected twice in the LC-MS/MS system. At 10 min the flow rate was diverted to the second column via a ten-port Valco valve (VICI Valco Instruments, Houston, TX, USA) in order to allow the injection of a second run on column 2 during the re-equilibration phase of column 1. During the first injection column 1 was used for analyte separation and column 2 was re-equilibrated; during the second injection column 2 was used for separation and column 1 was re-equilibrating (same condition). Data acquisition and elaboration were performed by the Analyst^®^ software (version 1.5.1, AB SCIEX).

### 4.3. Sample Collection and Storage

All the autopsies carried out by the forensic pathologists at the Department of Public Health (Experimental and Forensic Medicine, University of Pavia) between January 2017 and September 2018 were evaluated. Cardiac blood was collected only from cases where a recent intake of psychoactive substances was suspected due to the clinical records and circumstantial data (treatment with psychotropic drugs, overdoses, drug-related homicides, or suicides, etc.), including only the cases providing an adequate amount of sample (10 mL of blood). Overall, the method was applied to 60 post-mortem cases. For each case, blood samples together with DBSs were collected and stored as follows.

Whenever possible, at least 10 mL of cardiac blood were collected during autopsy (chosen instead of peripheral blood because the latter could be less abundant and are essential for forensic purposes), using a sterile syringe. Samples were stored in plastic tubes without the addition of any preservatives. 

Immediately after the collection, fifteen 85-µL aliquots of blood were pipetted onto three filter cards. After the drying process (in the dark, at room temperature, for at least two hours), the cards of each case were bagged in a paper envelope, without any desiccant, and kept in the same condition of shriveling over the whole study period. It was decided not to add any preservatives or desiccants in order to evaluate the reliability of the results from DBS analyses at the worst conditions of storage. The remaining blood collected in the plastic tube was refrigerated until the first analysis (performed within 24 h from the collection), then stored at −20 °C and thawed before the following analysis. 

Each sample was labelled and sealed with a proper code number. 

### 4.4. Sample Preparation

For each spot, the whole blood stain (an about 13-mm diameter disk) was cut and put into a glass tube containing 1 mL phosphate buffer solution at pH 6 as well as quetiapine-D8 (10 µL), clozapine-D4 (10 µL), and citalopram-D6 (10 µL) at the concentration of 100 ng/mL as internal standards. The solutions were sonicated for 10 min, vortexed for 10 s, and finally centrifuged at 4000 g for 5 min. Supernatant solutions were purified on a Bond Elut Certify I solid phase extraction (SPE, 200 mg) cartridge. The cartridges were initially activated with 2 mL methanol and then rinsed with 2 mL phosphate buffer solution at pH 6 before loading sample solutions. The columns were sequentially washed with 2 mL deionized water, 3 mL HCl 0.1 M, and finally 5 mL methanol. The analytes were then eluted from the column with 2 mL dichloromethane-isopropanol mixture (8:2 *v*/*v*) with 2% ammonia solution. The SPE eluate was dried under nitrogen stream and reconstituted in 200 µL mobile phase; finally, 5 µL were injected in the LC-MS/MS system. The same procedure was applied to 85 µL of blood samples stored in a freezer at −20° C. The method is summarized in Figure 20.

### 4.5. Stability Studies

For each case, DBSs were analyzed after drying process (time 0 = T0, within 24 h of collection).

A stability study was carried out over three months with time points of one week (T1), two weeks (T2), three weeks (T3), and three months (T4) of storage. For each time point, DBS samples were analyzed in triplicate.

The monitoring period was chosen based on the common time interval requested by prosecutors during criminal investigations to perform routine toxicological analyses and provide technical advice.

The blood stored in freezer was thawed and quantitatively analyzed at baseline (T0) and after three months (T4), and re-frozen after each analysis.

For cases 6 to 16, the blood stored in freezer was also analyzed at T1, T2, and T3 in order to better assess the stability of analytes even on conventional samples and to evaluate a possible effect of repeated thaw–freeze cycles. 

### 4.6. Validation Procedure

Limits of detection (LODs) were measured by evaluating the signal/noise (S/N) ratio of three replicates of spiked blank samples at the concentration of 10 ng/mL. The concentrations of spiked samples were increased for the analytes, providing low sensitivity. On the contrary, concentrations were decreased at 1 ng/mL for those analytes providing a high sensitivity (e.g., amisulpride). A peak with a S/N ≈ 3, adequate chromatography, and an acceptable ion ratio was selected as LOD level. Limits of quantification (LOQs) were fixed at administratively defined decision points and in accordance to their own sensitivity. 

The LOQs were calculated on ten fortified blank samples collected from different sources; all the blank samples were injected in duplicates and the detection, identification, bias, and precision criteria were evaluated. 

The method was fully validated only for the molecules detected in real samples. We optimized the instrumental parameters and we evaluated only selectivity and sensitivity for the remaining substances. Solid reference standards (salts) were prepared by dissolution of each analyte in methanol at the concentration of 1 mg/mL. Working solutions were freshly prepared in methanol at 6 different concentrations. Then, 15 µL of working solutions were added to calibration points in order to achieve a final concentration in blood of C1: 5/10 ng/mL (depending on LOQ); C2: 20 ng/mL; C3: 50 ng/mL; C4: 100 ng/mL; C5: 200 ng/mL; and C6: 500 ng/mL. A 1:10 dilution factor has been evaluated and accepted for quantitative determination. Concentrations exceeding 5000.0 ng/mL(/g) were reported as above the upper limit and were not considered in the statistics.

Quality control (QC) samples were prepared by a different operator by independent dilution at the concentration of 10 ng/mL, 20 ng/mL, 100 ng/mL, and 250 ng/mL. All standard solutions were stored at −20 °C until analysis. A total of 30 blank samples collected from different post-mortem cases were used for the method development, validation, and samples measurements (total volume stored ≈100 mL). Ten blank post-mortem samples were deposed on the paper substrate and analyzed for possibly interfering peaks during the first-step validation of the method. A methanolic solution containing more than 100 drugs among commonly prescribed drugs, drugs of abuse, and metabolites, at the final concentration of 1000 ng/mL, was added to samples. Selectivity was evaluated at two levels (LOQ and 500 ng/mL) and in blanks by adding 50 µL of a solution containing more than 100 substances, among drugs of abuse, cardiovascular agents, and psychoactive substances, before sample preparation. Linearity was verified by processing 10 calibration curves analyzed on 5 different days, over the whole range. Acceptance criteria included a coefficient of determination (r^2^) >0.99, and residuals within three standard deviations were considered adequate.

Intraday precision, expressed as coefficient of variation (CV%), was calculated analyzing the QC samples in five replicates, while inter-day precision was measured analyzing the QC samples in duplicate on five different days over a two-week period. The concentration of the analytes in the QC samples was calculated versus the daily calibration curves. Accuracy was determined as the error between the measured value at QC levels and the target concentration. Blank DBSs, blood samples (collected from five different sources), and water solutions were spiked at three levels (20, 50, and 250 ng/mL) before SPE extraction and after the purification; the absolute peak areas were compared in order to evaluate recovery and matrix effects, respectively. Experiments were carried out in quintuplicate. Recovery and matrix effects were expressed as the percentage of the mean deviation of drugs response in DBSs and in blood samples from the response measured in mobile phase at the same concentration level. Matrix effects were considered negligible whether the peak area ratios were within 20% variability. Recovery was expressed as the extraction efficiency percentage. Carry over was evaluated by means of injection of a blank sample after a blood sample fortified at the concentration of 1000 ng/mL and processed following the procedure described above.

## 5. Conclusions

We successfully developed an LC-MS/MS method for the identification of 20 antidepressants and 19 antipsychotics and quantification of 9 ADs and 5 APs in DBSs and blood. The method offered accurate quantification in terms of reliability and precision. The results provided a good qualitative and quantitative correlation between DBSs stored at room temperature and whole blood stored at −20 °C, except for fluvoxamine.

To the best of our knowledge, this study has been the first to evaluate the long-term stability of several antidepressants and antipsychotics in authentic post-mortem samples collected on DBS.

However, overall the outcomes from our study show that DBSs samples could represent a good complementary matrix to perform qualitative and quantitative analysis of ADs and APs.

The stability of most of the analytes in DBSs allows for a reliable quantitative result months after sample collection and storage at room temperature, even without the addition of any preservatives.

Though the analytical procedure must be applied to a greater population in order to provide statistically significant results, this is the first study applied on several authentic post-mortem cases, showing that DBSs could represent a good complementary sample storage in forensic investigations.

## Figures and Tables

**Figure 1 molecules-24-03636-f001:**
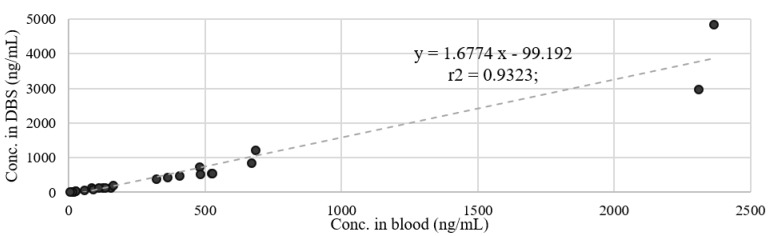
Comparison between cardiac blood and DBS concentrations in simultaneously collected specimens from 16 post-mortem cases.

**Figure 2 molecules-24-03636-f002:**
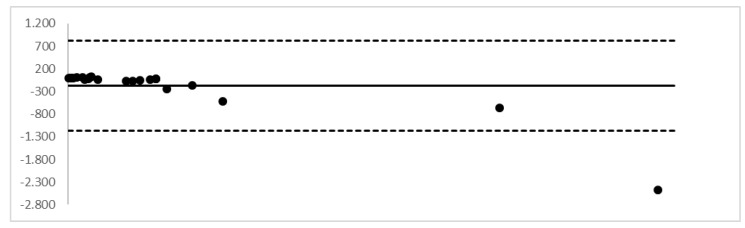
Bland–Altman plot. Evaluation of the agreement between DBSs and whole blood concentrations.

**Figure 3 molecules-24-03636-f003:**
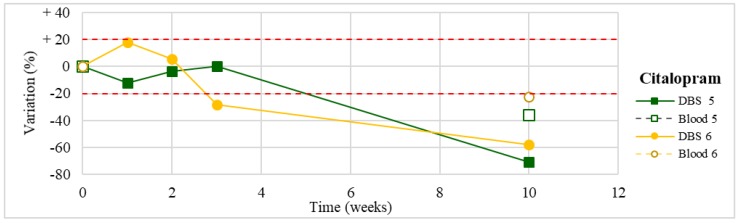
Long-term stability of citalopram (*n* = 2) in dried blood spots (DBSs) stored at room temperature and in blood stored at −20 °C (T0–T4).

**Figure 4 molecules-24-03636-f004:**
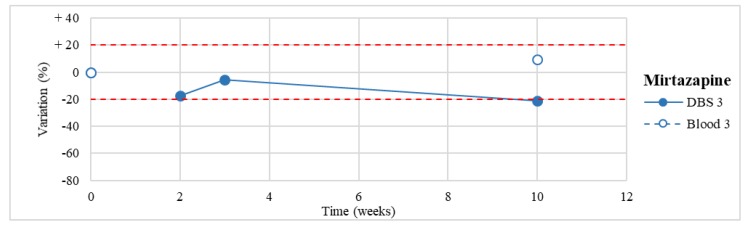
Long-term stability of mirtazapine (*n* = 1) in DBSs stored at room temperature and in blood stored at −20 °C (T0–T4).

**Figure 5 molecules-24-03636-f005:**
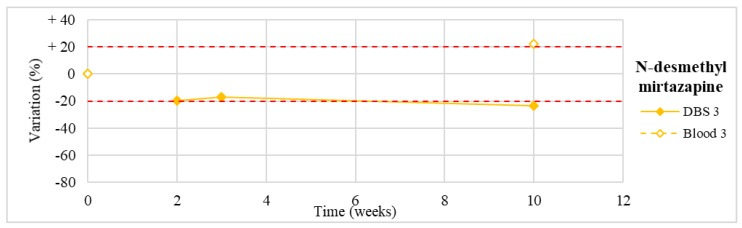
Long-term stability of N-desmethylmirtazapine (*n* = 1) in DBSs stored at room temperature and in blood stored at −20 °C (T0–T4).

**Figure 6 molecules-24-03636-f006:**
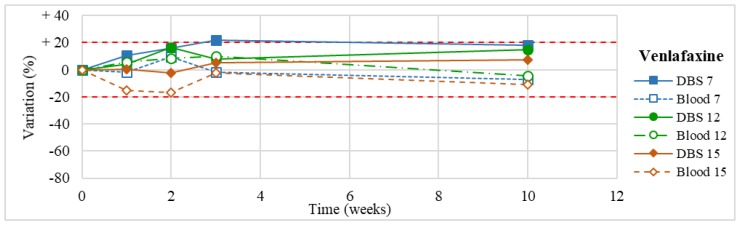
Long-term stability of venlafaxine (*n* = 3) in DBSs stored at room temperature and in blood stored at −20 °C (T0–T4).

**Figure 7 molecules-24-03636-f007:**
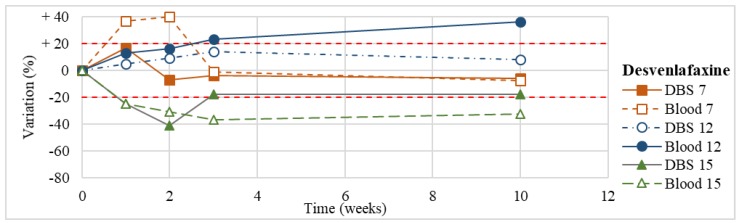
Long-term stability of desvenlafaxine (*n* = 3) in DBSs stored at room temperature and in blood stored at −20 °C.

**Figure 8 molecules-24-03636-f008:**
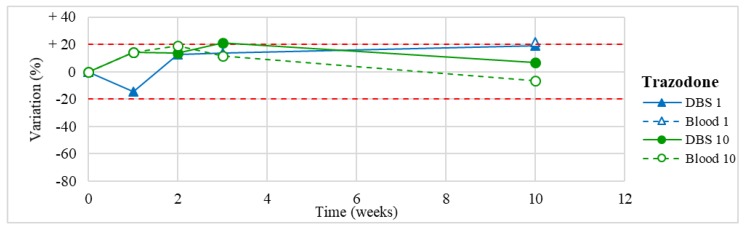
Long-term stability of trazodone (*n* = 2) in DBSs stored at room temperature and in blood stored at −20 °C (blood of case n°10 was tested also at T1, T2, T3).

**Figure 9 molecules-24-03636-f009:**
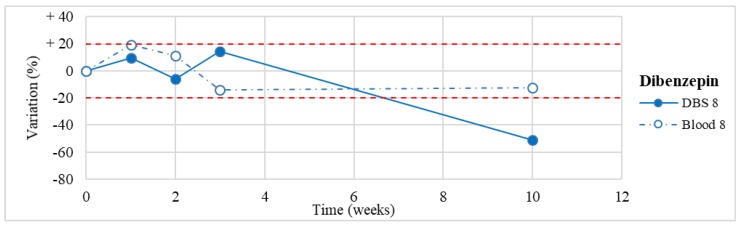
Long-term stability of dibenzepin (*n* = 1) in DBSs stored at room temperature and in blood stored at −20 °C.

**Figure 10 molecules-24-03636-f010:**
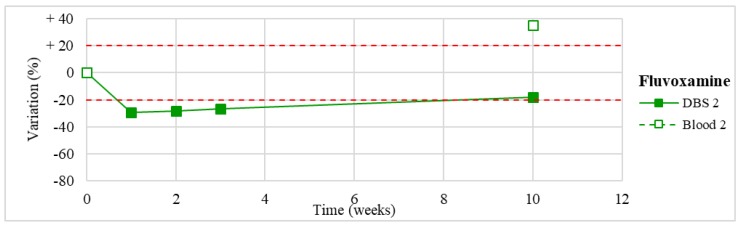
Long-term stability of fluvoxamine (*n* = 1) in DBSs stored at room temperature and in blood stored at −20 °C (T0–T4).

**Figure 11 molecules-24-03636-f011:**
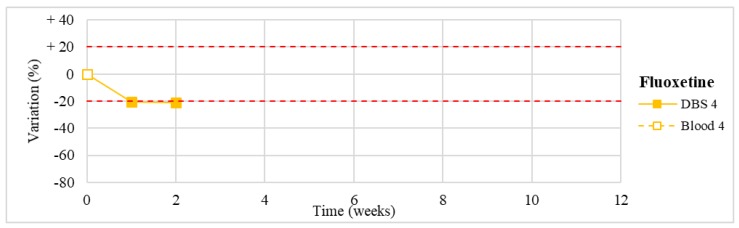
Stability of fluoxetine (*n* = 1) in DBSs stored at room temperature and in blood stored at −20 °C.

**Figure 12 molecules-24-03636-f012:**
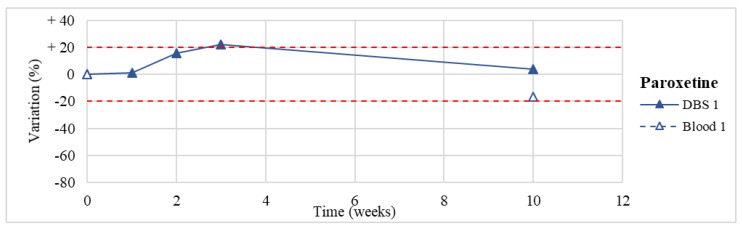
Long-term stability of paroxetine (*n* = 1) in DBSs stored at room temperature and in blood stored at −20 °C (T0–T4).

**Figure 13 molecules-24-03636-f013:**
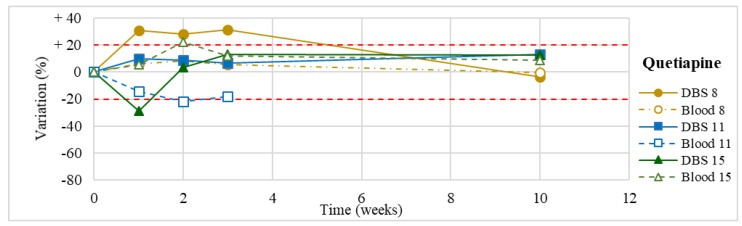
Long-term stability of quetiapine (*n* = 3) in DBSs stored at room temperature and in blood stored at −20 °C.

**Figure 14 molecules-24-03636-f014:**
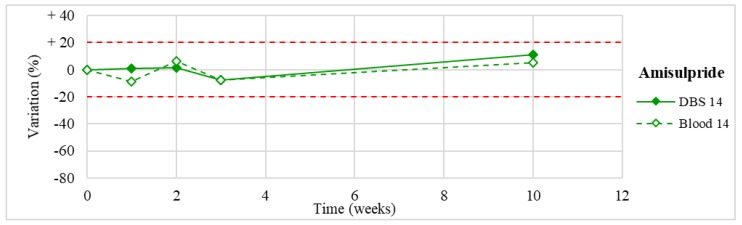
Long-term stability of amisulpride (*n* = 1) in DBSs stored at room temperature and in blood stored at −20 °C.

**Figure 15 molecules-24-03636-f015:**
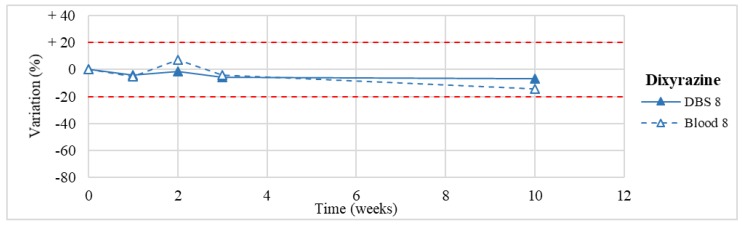
Long-term stability of dixyrazine (*n* = 1) in DBSs stored at room temperature and in blood stored at −20 °C.

**Figure 16 molecules-24-03636-f016:**
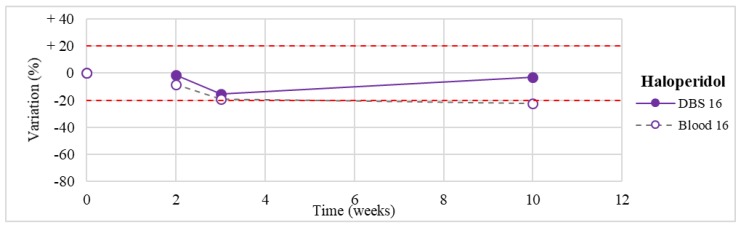
Long-term stability of haloperidol (*n* = 1) in DBSs stored at room temperature and in blood stored at −20 °C.

**Figure 17 molecules-24-03636-f017:**
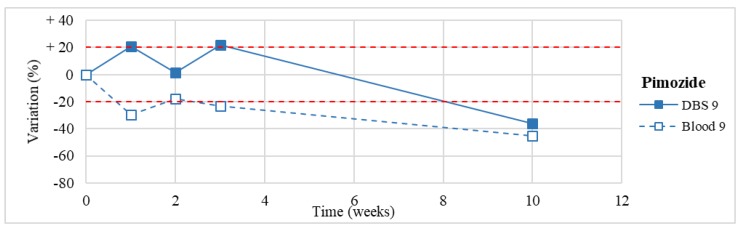
Long-term stability of pimozide (*n* = 1) in DBSs stored at room temperature and in blood stored at −20 °C.

**Figure 18 molecules-24-03636-f018:**
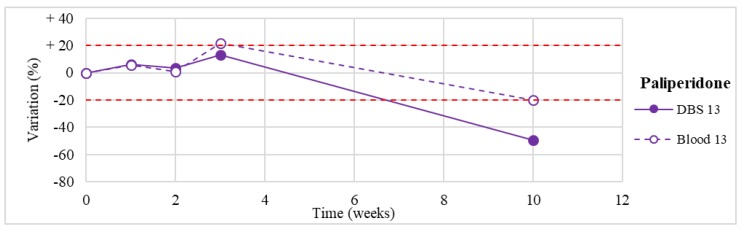
Long-term stability of paliperidone (*n* = 1) in DBSs stored at room temperature and in blood stored at −20 °C.

**Figure 19 molecules-24-03636-f019:**
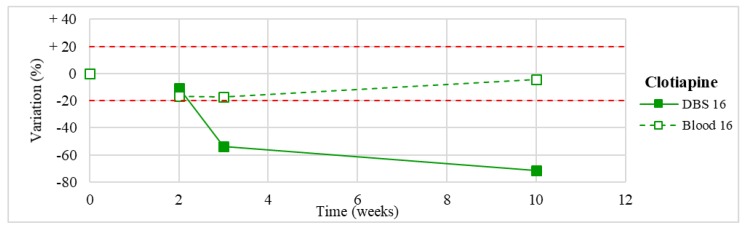
Long-term stability of clotiapine (*n* = 1) in DBS stored at room temperature and in blood stored at −20 °C.

**Figure 20 molecules-24-03636-f020:**
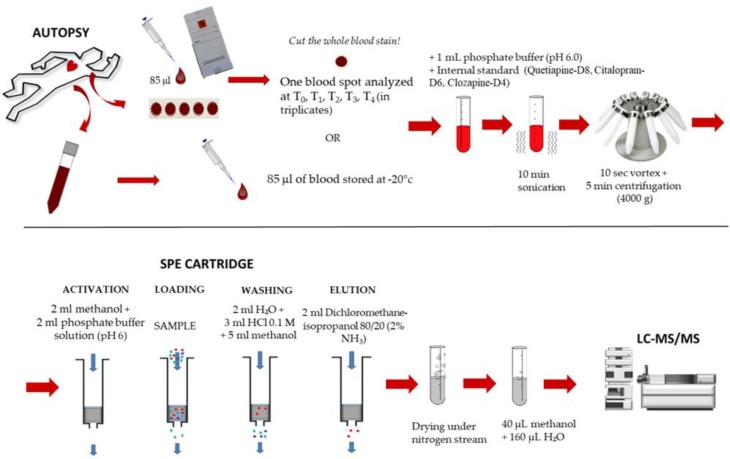
Sample preparation and extraction.

**Table 1 molecules-24-03636-t001:** Limits of detection (LODs) and quantification (LOQs).

Substance Antidepressants	LOD (ng/mL)	LOQ (ng/mL)	Substance Antipsychotics	LOD (ng/mL)	LOQ (ng/mL)
Amitriptyline	0.6	/	Amisulpride	0.2	5.0
Citalopram	2.3	5.0	Asenapine	4.1	/
Desipramine	0.9	/	Chlorpromazine	0.4	/
N-Desmethyl-mirtazapine	1.4	5.0	Clotiapine	1.8	10.0
Desvenlafaxine	3.2	5.0	Clozapine	1.4	/
Dibenzepin	0.3	5.0	Dixyrazine	1.3	5.0
Dothiepin	1.8	/	Duloxetine	1.3	/
Fluoxetine	1.6	5.0	Fluphenazine	0.3	/
Fluvoxamine	1.9	5.0	Haloperidol	0.8	10.0
Maprotiline	0.6	/	Hydroxyzine	0.9	/
Mianserin	1.0	/	Levomepromazine	5.2	/
Mirtazapine	2.2	10.0	Olanzapine	2.7	/
Nortriptyline	0.5	/	Paliperidone	1.4	10.0
Paroxetine	0.6	/	Pimozide	2.1	5.0
Protriptyline	2.0	/	Promazine	0.1	/
Reboxetine	1.1	/	Promethazine	2.0	/
Sertraline	0.5	/	Quetiapine	0.8	10.0
Trazodone	0.5	5.0	Tiapride	0.4	/
Trimipramine	0.1	5.0	Ziprasidone	1.8	/
Venlafaxine	0.1	5.0			

**Table 2 molecules-24-03636-t002:** Accuracy and precisions (5 replicates for each measurement). CV%: coefficient of variation.

Substance	Accuracy (bias%)					Intra-Day Precision (CV%)					Inter-Day Precision (CV%)				
	5 ng/mL	10 ng/mL	20 ng/mL	100 ng/mL	250 ng/mL	5 ng/mL	10 ng/mL	20 ng/mL	100 ng/mL	250 ng/mL	5 ng/mL	10 ng/mL	20 ng/mL	100 ng/mL	250 ng/mL
Amisulpride	6.6	2.2	0.2	0.9	1.3	20.0	5.3	8.6	6.6	8.2	15.4	20.0	8.0	16.9	2.1
Clotiapine	/	7.0	11.5	3.6	0.4	/	3.7	7.1	16.3	6.9	/	17.1	17.4	12.5	4.5
Haloperidol	/	7.7	4.7	2.1	0.2	/	11.3	7.9	14.6	12.3	/	18.9	2.2	17.2	14.1
Paliperidone	/	6.8	6.1	3.0	0.4	/	19.5	8.5	15.0	19.3	/	17.9	7.3	5.7	5.7
Quetiapine	/	8.1	1.0	1.0	0.2	/	19.5	17.2	6.2	5.3	/	2.8	15.9	5.2	17.9
Citalopram	4.5	1.6	4.0	1.8	0.3	9.3	18.2	4.9	9.7	4.2	4.5	18.6	13.2	7.6	8.1
Desvenlafaxine	5.9	0.3	2.7	0.9	0.1	17.5	18.4	14.2	8.9	15.6	9.8	18.8	12.7	12.8	5.6
Dibenzepin	5.6	3.8	1.3	0.4	1.6	19.8	20.0	11.0	18.1	2.2	7.2	6.3	8.4	14.8	1.2
Fluoxetine	16.6	0.4	0.8	1.5	0.3	18.6	6.2	17.6	6.5	7.1	19.6	17.4	14.7	16.1	3.1
Fluvoxamine	0.6	8.5	3.5	2.8	0.3	20.0	2.5	16.0	11.9	18.9	18.6	17.6	16.9	20.0	20,0
Mirtazapine	/	10.5	1.6	3.0	0.5	/	15.9	11.9	18.6	7.0	/	20.0	10.0	10.2	5.9
N-Desmethyl-mirtazapine	0.5	6.3	2.1	0.3	2.3	19.6	18.0	4.1	2.7	18.2	6.2	16.5	8.4	18.6	3.6
Trazodone	12.9	2.0	0.6	1.7	0.1	18.9	19.9	15.6	15.3	5.5	4.9	20.0	5.7	9.6	7.8
Venlafaxine	6.0	3.4	5.3	1.2	0.2	17.9	3.5	11.5	12.2	6.9	10.8	2.6	8.3	15.3	2.2

**Table 3 molecules-24-03636-t003:** Recovery and matrix effects results (5 replicates for each measurement).

Substance	Recovery %				Matrix Effects %			
	5 ng/mL	10 ng/mL	20 ng/mL	250 ng/mL	5 ng/mL	10 ng/mL	20 ng/mL	250 ng/mL
Amisulpride	115.7	110.9	79,6	103.9	+16.2	+11.8	+19.3	+4.1
Clotiapine	/	119.2	118.8	113.5	/	+20.3	−17.8	−7.7
Haloperidol	/	114.1	70.5	103.1	/	−12.2	−19.1	+14.5
Paliperidone	/	53.4	59.4	44.7	/	+17.6	−6.0	+3.9
Quetiapine	/	50.4	40.8	26.9	/	+5.5	+16.2	+17.8
Citalopram	99.2	98.5	66.9	92.1	+1.5	−12.9	−12.8	−1.9
Desvenlafaxine	95.2	85.4	85.4	89.3	−2.1	−11.5	+13.9	−11.6
Dibenzepin	117.8	106.4	76.5	103.9	−10.4	+19.7	−4.3	+20.5
Fluoxetine	120.0	49.7	32.1	57.0	−16.3	−0.7	+19.4	+18.5
Fluvoxamine	118.8	69.2	90.1	86.6	−1.0	−16.8	−4.1	+20.6
Mirtazapine	/	116.7	77.0	101.6	/	−0.6	−18.5	+4.6
N-desmethyl-mirtazapine	102.4	117.6	87.1	87.3	−18.6	−18.5	−16.5	−3.6
Trazodone	88.0	112.4	59.8	95.9	−2.2	−19.4	−19.6	+20.6
Venlafaxine	116.4	118.7	87.3	94.4	+0.6	−10.2	−13.7	−13.2

**Table 4 molecules-24-03636-t004:** Concentrations of the 16 positive dried blood spot (DBS) samples at T0.

Case Number	Substance	Blood Therapeutic Range Used in the Laboratory (ng/mL)	Concentration in DBS (ng/mL)
**1**	Trazodone Paroxetine	700.0–1000.0 10.0–50.0	91.1 671.6
**2**	Fluvoxamine	60.0–230.0	2366.0
**3**	Mirtazapine N-desmethylmirtazapine	30.0–80.0 /	133.7 112.1
**4**	Fluoxetine	120.0–500.0	5.6
**5**	Citalopram	50.0–110.0	408.3
**6**	Citalopram	50.0–110.0	56.9
**7**	Venlafaxine Desvenlafaxine	100.0–400.0 100.0–400.0	362.5 523.5
**8**	Dibenzepin Quetiapine Dixyrazine	25.0–150.0 100.0–500.0 ≈300.0	483.4 2309.7 126.4
**9**	Pimozide	4.0–10.0	13.1
**10**	Trazodone	700.0–1000.0	162.0
**11**	Quetiapine	100.0–500.0	84.1
**12**	Venlafaxine Desvenlafaxine	100.0–400.0 100.0–400.0	25.6 479.5
**13**	Paliperidone	20.0–60.0	17.6
**14**	Amisulpride	100.0–400.0	527.2
**15**	Venlafaxine Desvenlafaxine Quetiapine	100.0–400.0 100.0–400.0 100.0–500.0	321.7 686.1 153.5
**16**	Clotiapine Haloperidol	5.0–170.0 5.0–40.0	24.5 19.0

**Table 5 molecules-24-03636-t005:** Comparison between our stability results and those ones obtained from previously published papers.

Analytes (Number of Positive Cases)	Stability of the Analytes on DBSs Observed in the Present Study	Stability of the Analytes in Similar Matrices According to Previously Published Articles
Citalopram (*n* = 2)	Stable for the first 2 weeks in one case and for 3 weeks in the other one. Degradation >50% after 3-month storage.	Karinen et al. [76]: stable for up to 1 year in authentic post-mortem blood samples after storage at −20 °C. Lewis et al. [77]: stable for 5 days in whole blood specimens stored at 4 °C.
Mirtazapine (*n* = 1)	Stable within the 3-month period.	Lavasani et al. [78]: stable in plasma samples kept at −20 °C for at least 6 months. Kuchecar et al. [79]: stable respectively for 12 h and 58 h at room temperature on bench top and in autosampler.
N-desmethyl mirtazapine (*n* = 1)	Stable for the first 3 weeks, with a slight degradation (−24%) at T4.	
Venlafaxine (*n* = 3)	Stable within the 3-month period.	Butzbach et al. [80]: stable in aqueous solutions and biological matrixes for 57 days at 20 °C. Berm et al. [44]: stable in spiked blood on DBS cards within a period of 6 months of storage at 2–8 °C.
Desvenlafaxine (*n* = 3)	Stable within the 3-month period (except for an apparent degradation at T1 in one case, not confirmed at further analyses).	
Trazodone (*n* = 1)	Stable within the 3-month period.	/
Dibenzepin (*n* = 1)	Stable for the first 3 weeks. Degradation >50% after 3-month storage on DBSs (but not in blood stored at −20° C).	/
Fluvoxamine (*n* = 1)	Moderate degradation in the first week (about −30%), remaining substantially stable afterward.	/
Fluoxetine ^§^ (*n* = 1)	Slight degradation after the first week of storage (about −20%), remaining stable at the further analyses.	Lantz et al. [65]: stable in plasma for up to 96 h at room temperature and up to one year at −20 °C. Binsumait et al. [81]: significant loss in plasma at the second, third, and fifth weeks of storage at room temperature. Karinen et al. [82]: significant loss in stock solutions at room temperature after one year (−43.1%).
Paroxetine * (*n* = 1)	Stable within the 3-month period.	Déglon et al. [42]: stable on DBSs (obtained from spiked blood) for 1 month at room temperature and at 40 °C.
Dixyrazine * (*n* = 1)	Stable within the 3-month period.	/
Quetiapine (*n* = 3)	In one case, moderate increase in the first 3 weeks (<31%). In the other two cases, stable within the 3-month period (except for an apparent degradation of 29% at T1 in one of them, not confirmed at further analyses).	Heller et al. [60]: stable in plasma samples at room temperature for at least 7 days and at −20 °C for 3 months. Saar et al. [52]: stable in blank blood spiked with drugs, stored at different temperatures (20 °C, 4 °C, −20 °C, and −60 °C) for 10 weeks. Youssef et al. [83]: stable in plasma and spiked whole blood samples after 20 weeks of storage at room temperature (≤20% of degradation).
Amisulpride (*n* = 1)	Stable within the 3-month period.	
Haloperidol (*n* = 1)	Stable within the 3-month period.	/
Pimozide * (*n* = 1)	Stable for the first 3 weeks. Degradation >50% after 3-month storage.	/
Paliperidone (*n* = 1)	Stable for the first 3 weeks. Degradation >50% after 3-month storage on DBSs (but not in blood stored at −20° C).	Butzbach et al. [84]: stable in sterile porcine blood at 7, 20, and 37 °C for 4 days. Unstable at the same storage conditions in bacterially inoculated porcine blood.
Clotiapine (*n* = 1)	Unstable after 2 weeks, with >70% loss after 3 months.	/

* Analyte that did not fulfil all the criteria for validation; ^§^ monitored only for 2 weeks.

**Table 6 molecules-24-03636-t006:** Chromatographic conditions.

**Liquid Chromatography**	
Flow rate	200 µL/min → 400 µL/min constant flow of 0.2 mL/min; gradient elution: 90% A maintained for 2.5 min, then from 90% to 10% A within 3.0 min, maintaining 10% A up to 10.0 min, and re-equilibration up to 20 min.
Mobile phase	H_2_O 0.1% (*v*/*v*) formic acid (A) Acetonitrile 0.1% (*v*/*v*) formic acid (B)
Type of elution	gradient
Column	Kinetex C18 (100 × 2,1 mm, 5 µm particle size) (Phenomenex, Castelmaggiore, BO, Italy) Kept at 45° during the analysis
**Mass Spectrometry**	
Operative mode	Multiple reaction monitoring (MRM)—positive polarity using nitrogen as collision gas (pressure set at level 5). Dwell time: 30 ms.
Ion spray voltage	5000 V
Source temperature	350 °C
Curtain gas	15 PSI
Nebulization gas (air)	20 PSI
Heating gas (air)	25 PSI

**Table 7 molecules-24-03636-t007:** Multiple reaction monitoring (MRM) transitions for each substance. Quantifier transitions in bold.

ANALYTE	Q1 (*m*/*z*)	Q3 (*m*/*z*)	DP *(V)	EP *(V)	CE *(V)	CXP *(V)
Antidepressants						
Amitriptyline	**278.1**	**233.3**	**102**	**10**	**25**	**5**
	278.1	91.2	102	10	35	3
Citalopram	**325.1**	**109.2**	**100**	**9**	**35**	**4**
	325.1	262.3	100	9	28	6
Citalopram-D6	**331.1**	**109.1**	**85**	**10**	**37**	**5**
	331.1	262.1	85	10	28	10
Desipramine	**266.9**	**72.2**	**71**	**10**	**28**	**11**
	266.9	208.3	71	10	33	10
N-Desmethylmirtazapine	**252.0**	**195.2**	**95**	**9**	**32**	**8**
	252.0	209.1	95	9	32	9
Desvenlafaxine	**265.0**	**58.1**	**62**	**10**	**48**	**7**
	265.0	202.0	62	10	25	8
Dibenzepin	**295.9**	**250.9**	**90**	**10**	**35**	**10**
	295.9	209.2	90	10	47	10
Dothiepin	**295.9**	**223.2**	**75**	**8**	**33**	**11**
	295.9	218.3	75	8	33	11
Fluoxetine	**310.1**	**310.1**	**63**	**8**	**7**	**10**
	310.1	148.3	63	8	13	7
Fluvoxamine	**319.0**	**258.3**	**69**	**8**	**16**	**7**
	319.0	71.2	69	8	29	11
Maprotiline	**278.2**	**250.3**	**148**	**8**	**28**	**13**
	278.2	219.3	148	8	36	11
Mianserin	**264.9**	**208.3**	**103**	**12**	**30**	**10**
	264.9	58.3	103	12	47	8
Mirtazapine	**267.0**	**196.0**	**100**	**10**	**35**	**8**
	267.0	72.0	100	10	32	9
Nortriptyline	**263.9**	**233.3**	**80**	**9**	**22**	**12**
	263.9	91.2	80	9	35	3
Paroxetine	**330.4**	**192.3**	**144**	**9**	**30**	**9**
	330.4	70.2	144	9	49	11
Protriptyline	**264.9**	**192.2**	**110**	**10**	**34**	**7**
	264.9	156.2	110	10	29	7
Reboxetine	**313.8**	**176.0**	**81**	**10**	**31**	**29**
	313.8	91.1	81	10	43	3
Sertraline	**306.0**	**159.2**	**63**	**8**	**37**	**7**
	306.0	275.2	63	8	18	6
Trazodone	**372.2**	**176.3**	**97**	**9**	**35**	**8**
	372.2	148.2	97	9	48	6
Trimipramine	**295.0**	**100.3**	**69**	**10**	**23**	**4**
	295.0	58.1	69	10	59	8
Venlafaxine	**278.0**	**58.2**	**71**	**10**	**40**	**6**
	278.0	121.1	71	10	40	3
**Antipsychotics**						
Amisulpride	**369.9**	**242.2**	**101**	**9**	**39**	**12**
	369.9	112.5	101	9	39	4
Asenapine	**286.2**	**165.9**	**96**	**10**	**46**	**6**
	286.2	215.0	96	10	40	8
Chlorpromazine	**318.8**	**86.2**	**78**	**7**	**29**	**3**
	318.8	246.2	78	7	34	12
Clotiapine	**343.9**	**287.2**	**101**	**14**	**30**	**7**
	343.9	255.3	101	14	44	5
Clozapine	**327.0**	**270.1**	**100**	**9**	**34**	**6**
	327.0	296.3	100	9	36	7
Clozapine-D4	**331.3**	**272.3**	**94**	**10**	**35**	**9**
	331.3	299.1	94	10	35	11
Dixyrazine	**427.9**	**229.3**	**104**	**10**	**37**	**11**
	427.9	187.3	104	10	39	9
Duloxetine	**298.2**	**154.1**	**38**	**8**	**8**	**5**
	298.2	188.0	38	8	8	7
Fluphenazine	**437.9**	**171.3**	**109**	**10**	**38**	**8**
	437.9	143.3	109	10	45	6
Haloperidol	**375.9**	**165.2**	**83**	**10**	**34**	**7**
	375.9	123.1	83	10	57	5
Hydroxyzine	**376.1**	**202.2**	**64**	**10**	**26**	**10**
	376.1	166.2	64	10	70	7
Levomepromazine	**330.0**	**100.3**	**72**	**14**	**30**	**4**
	330.0	243.2	72	14	33	12
Olanzapine	**330.0**	**100.3**	**72**	**14**	**30**	**4**
	330.0	243.2	72	14	33	12
Paliperidone	**427.3**	**207.2**	**122**	**10**	**40**	**8**
	427.3	110.1	122	10	60	2
Pimozide	**462.1**	**328.4**	**126**	**10**	**43**	**8**
	462.1	109.2	126	10	88	4
Promazine	**284.8**	**86.2**	**65**	**12**	**29**	**13**
	284.8	212.3	65	12	35	10
Promethazine	**285.0**	**198.0**	**80**	**10**	**18**	**15**
	285.0	240.0	80	10	20	15
Quetiapine	**384.5**	**253.5**	**96**	**10**	**31**	**6**
	384.5	279.1	96	10	34	7
Quetiapine-D8	**392.3**	**258.3**	**90**	**10**	**35**	**12**
	392.3	286.5	90	10	34	15
Tiapride	**328.9**	**256.0**	**118**	**6**	**26**	**20**
	328.9	213.2	118	6	48	10
Ziprasidone	**413.3**	**194.2**	**121**	**10**	**39**	**9**
	413.3	177.3	121	10	39	8

* DP: declustering potential; EP: entrance potential; CE: collision energy; CXP: collision cell exit potential.

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
