# Peer review of "Determination of Antidepressants and Antipsychotics in Dried Blood Spots (DBSs) Collected from Post-Mortem Samples and Evaluation of the Stability over a Three-Month Period"

_molecules, 2019, doi:10.3390/molecules24203636_

Round 1

Reviewer 1 Report

The manuscript refers to the determination of 22 antidepressants and 19 antipsychotics in dried blood spots and whole blood samples by LC-MS/MS. The authors stated at the introduction that this is the first reported application in real post-mortem DBS samples. The research theme and the experimental work presented in this manuscript are interesting and to me this work deserves publication after revision.

Comments:

The overall quality of the paper regarding readability is poor; in addition the manuscript should be better organized in accordance to Molecules guidelines and the number of Figures should be reduced. The authors should further explain the novelty and the implementation of the proposed method in the introduction Table 1. Q1 (m/z) + Q3 (m/z), use 1 decimal; the terms DP (V), EP (V), CE (V) and CXP should be explained In Line 206, in table 3 and all over the test, please replace the term imprecision …. By “precision”, Table 2: indicate decimal points by full stops Fig2 to Fig 19, reduce the number of figures and describe the conclusions within the text

Reviewer 2 Report

The topic of this manuscript is interesting. It can be accepted after some amendments.

(1) How to demonstrate the selectivity of the assay?

(2) The MS ion pairs (m/z) used in the assay may have been applied before. The references should be quoted.

(3) If the results obtained from DBS are so different from the blood, how to convence authority to use DBS? 

(4) The stability of analytes in blood under frozen condition appears to be better, right? So, what is the value of DBS?

(5) As the reviewer is not a forensic scientist, the reviewer wants to know if the authors validated the method following forensic guidance. 

Reviewer 3 Report

In this manuscript the authors sought to develop a LC-MS/MS method for the identification and quantification of antidepressants and antipsychotics on dried blood spot (DBS). In view of this study, the authors showed that a whole blood stain was extracted in phosphate buffer and purified using SPE cartridges for analyzing those molecules. Overall it is a passably written manuscript that it is the analytical procedure could be applied on a greater population in order to provide statistically significant results, this is a study applied on several authentic post-mortem cases, showing that DBS could represent a complementary sample storage in forensic investigations. However, there are some points the need further clarification as follows.

Abstract: this part is insufficient, especially in matrix effect. It is need to be supplementary. Introduction: This part is tedious and hard to follow. These description are occupied too much spaces; these authors should be written more concisely. Results: The authors need to express that low recovery ratio at Paliperidone, Quetiapine, and etc. These lower recovery ratio of analyzing drugs could not be considerable at here. Discussion: The discussion need more information to show the significant among different studies or references, especially for multi-analysis in antidepressants and antipsychotics Conclusions: these descriptions of this part is not different with Introduction or Disscussion. I am not sure the authors wanted to show the “Introduction”, “discussion” or “Conclusions”.
